# Attention-Based Batch Normalization for Binary Neural Networks

**DOI:** 10.3390/e27060645

**Published:** 2025-06-17

**Authors:** Shan Gu, Guoyin Zhang, Chengwei Jia, Yanxia Wu

**Affiliations:** College of Computer Science and Technology, Harbin Engineering University, Harbin 150009, China

**Keywords:** Binary neural networks, batch normalizationa, deep learning, convolutional neural networks

## Abstract

Batch normalization (BN) is crucial for achieving state-of-the-art binary neural networks (BNNs). Unlike full-precision neural networks, BNNs restrict activations to discrete values {−1,1}, which requires a renewed understanding and research of the role and significance of the BN layers in BNNs. Many studies notice this phenomenon and try to explain it. Inspired by these studies, we introduce the self-attention mechanism into BN and propose a novel Attention-Based Batch Normalization (ABN) for Binary Neural Networks. Also, we present an ablation study of parameter trade-offs in ABN, as well as an experimental analysis of the effect of ABN on BNNs. Experimental analyses show that our ABN method helps to capture image features, provide additional activation-like functions, and increase the imbalance of the activation distribution, and these features help to improve the performance of BNNs. Furthermore, we conduct image classification experiments over the CIFAR10, CIFAR100, and TinyImageNet datasets using BinaryNet and ResNet-18 network structures. The experimental results demonstrate that our ABN consistently outperforms the baseline BN across various benchmark datasets and models in terms of image classification accuracy. In addition, ABN exhibits less variance on the CIFAR datasets, which suggests that ABN can improve the stability and reliability of models.

## 1. Introduction

Binary neural networks (BNNs) are one of the recent directions in deep neural networks (DNNs) edge implementations, which are proposed to save energy and improve computational efficiency [1,2,3,4,5]. BNNs, as an extreme form of quantized neural networks, require only 1 bit of two discrete values {−1,+1} to represent the network’s weights and/or activations. Since the 1-bit convolution operation can be efficiently implemented by XNOR and Bitcount operations, BNNs can significantly speed up and save energy in inference. However, such extreme constraints often lead to underfitting, making BNNs simultaneously disadvantageous in terms of training difficulties and accuracy degradation.

Batch Normalization (BN) [6] is a widely used technique in deep learning and most of BNNs contain BN layers because they are infeasible to train without BN layers [7]. Unlike full-precision neural networks, BNNs restrict activations to discrete values {−1,+1}, so that the forward pass is not affected by the scale of the input distribution, but only by the location of the distribution. For BNNs, scaling in BN layers prevents gradient explosion, while bias in BN layers determines the choice of activation values between 1 and −1, which is very different from full-precision neural networks. In addition, BNNs usually use the sign function as the activation function, which has insufficient nonlinear fitting ability and is not differentiable. Although back-propagation is possible using gradient approximation methods, the difference between the actual function and the approximated function can lead to gradient mismatch problems. These challenges create opportunities for optimizing the BN layer in BNNs, providing an impetus for BN research and improvements targeting BNNs.

In this work, we introduce the self-attention mechanism into the BN layer in BNNs by proposing Attention-based Batch Normalization for BNNs (ABN), a lightweight and input-adaptive normalization method specifically designed for BNNs.Due to the introduction of the self-attention mechanism, the ABN method has a more significant ability to highlight the unique features of each category compared to the baseline BN method. And, ABN enhances the ability of the network to capture complex patterns and nonlinear relationships in the data from the point of view of additional activation functions due to the use of sigmoid functions in the self-attention mechanism. In addition, in our experiments, we observe that our ABN approach further increases the imbalance of the binary activation distribution compared to the baseline BN method. This allows BNNs to exploit the diversity of data more effectively, enhance the model’s generalization ability, and improve the model’s performance on various tasks.

**Contributions.** In summary, our contributions are as follows:We propose a lightweight attention-based batch normalization method tailored for Binary Neural Networks (BNNs), named Attention-based Batch Normalization (ABN). While prior works have explored attention mechanisms within BN, ABN is, to our knowledge, the first to replace the learnable scaling factor γ with a parameter-free, input-dependent attention function specifically designed for binary activations.ABN rescales the normalized activations using a sigmoid-based self-attention mechanism derived directly from the input, which enhances the network’s ability to capture complex patterns and nonlinear relationships, increases the imbalance of binary activation distributions, and improves generalization.ABN is a plug-and-play module that can directly replace standard BN in binary activation networks, and it can be easily integrated into existing binary architectures without modifying the overall design.Extensive experiments demonstrate that ABN consistently outperforms standard BN across three benchmark datasets and two commonly used network backbones.

## 2. Related Work

### 2.1. Batch Normalization in BNNs

Batch Normalization (BN) [6] is a widely used technique in deep learning that helps to train DNNs faster and more consistently [8,9]. Different normalization variants differ in how they partition input data (e.g., by instances [10], channels [11], groups [12], positions [13], and image domains [14,15]). These variants typically have the same learnable linear transformation module, and the output is not constrained to obey zero-mean and unit-variance distributions, improving the ability to fit real-world data distributions.

In the realm of full-precision networks, it is widely acknowledged that BN operates by aligning the first and second moments of input and output distributions. This correction process effectively addresses covariate bias and restrains the gradient, ultimately leading to a smoother optimization process [8]. BN layers exhibit different effects in Binary neural networks (BNNs) compared to full-precision networks For training BNNs to converge, BN is essential, and as argued by Sari et al., it plays a crucial role in preventing gradient explosion [7].

In the case of BNNs employing sign activation, it’s worth noting that the scale of the input distribution does not impact the forward pass; rather, it’s the bias that assumes a pivotal role. According to Sari et al.’s argument, when inference is executed, the BN layers followed by the sign function effectively function as a thresholding mechanism [7]. Furthermore, the magnitude of the bias within BN layers holds a significant influence over the resulting activation values.

Recent studies have explored incorporating attention mechanisms into batch normalization to improve its adaptability. For instance, Liang et al. proposed Instance Enhancement Batch Normalization (IEBN) [16], which uses learnable instance-specific modulation to handle noisy data, while Martinez et al. [17] applied a channel-wise attention mechanism via a multi-layer gating structure, achieving improved performance in BNNs. However, these methods often introduce additional parameters and increase network complexity. Meanwhile, several works have focused on optimizing or replacing BN for better efficiency in BNNs. Chen et al. [18] proposed a BN-free training scheme using gradient clipping and weight standardization. Vorabbi et al. [19] introduced BNN-Clip to reduce BN-induced latency and data width, and Rege et al. [20] designed an in-memory BN method for hardware-efficient BNN deployment. These developments highlight the need for lightweight and adaptive normalization, motivating our attention-based BN approach designed specifically for BNNs.

### 2.2. Unbalanced Activation Distribution in BNNs

Given that BNNs only have two activation values (+1 and −1), the distribution of binary activations has been demonstrated to be a critical factor in their performance. The main focus of Ding et al.’s [21] work was to regulate the distribution of pre-activation values, with a particular emphasis on avoiding extreme cases where all pre-activation values share the same sign. Liu et al. [5] suggested balancing the distribution of binary activation by reshaping it with trainable thresholds and activation functions.

After conducting extensive experimental analysis, Kim et al. [22] come to the conclusion that unbalanced activation distributions can, in fact, enhance the accuracy of neural networks. As is widely recognized, most DNN models tend to employ the ReLU activation function as opposed to sigmoid or Tanh functions. The output distributions of the sigmoid and Tanh functions are symmetric with respect to zero, whereas the ReLU function replaces all negative values with zero, resulting in a highly skewed output distribution. However, when binarized activations are utilized, the sign function is employed instead, which leads to a symmetric distribution of binarized activations. Therefore, it was hypothesized that the symmetry of the sign function may contribute to the degradation of BNN performance.

### 2.3. Self-Attention Mechanism

The advantages of the self-attention mechanism include better capture of important information in the input vectors [23,24,25], smooth nonlinear properties [26], better handling of nonlinear relationships in the input vectors [26], etc., which will help to improve the underfitting of the BNNs due to extreme constraints. The sigmoid function is a widely employed tool in attention mechanisms for calculating attention weights. It is utilized to transform the input attention score into a normalized range, specifically between 0 and 1, thereby representing the significance of each pixel. In addition, the incorporation of the sigmoid function in the self-attention mechanism introduces a valuable nonlinear transformation that significantly enhances the model’s ability to capture intricate relationships within an image.

Binary neural networks lose a large amount of information during training due to their binary nature and also have limited nonlinear fitting ability due to their usual use of symbolic functions as activation functions [27]. Incorporating the self-attention mechanism into BN layers of BNNs brings about dual advantages. Firstly, by combining the self-attention mechanism with the BN layer, BNNs can benefit from both the attention-important feature of self-attention and the standardization property of BN, thus enhancing the representation capability of the network. Secondly, the presence of the self-attention mechanism with its inherent nonlinear function enhances the network’s ability to perform nonlinear fitting, enabling it to model complex relationships in the data.

## 3. Preliminaries

### 3.1. STE-Based Binary Neural Networks

The main operation in deep neural networks is expressed as:(1)$$z=w^{⊤}a$$
where $w\in R^{n}$ indicates the weight vector, $a\in R^{n}$ indicates the input activation vector computed by the previous network layer.

The goal of network binarization is to represent the floating-point weights and/or activations with 1-bit. And we usually use sign function to get $B_{x}$:(2)$$B_{x}=sign\left(x\right)=\left\{\begin{array}{cc} +1, & if\;x\geq 0\\ -1, & otherwise \end{array}\right.$$

With the quantized weights and activations, the vector multiplications in the forward propagation can be reformulated as(3)$$z=B_{w}⊙B_{a}$$
where ⊙ denotes the inner product for vectors with bitwise operations XNOR and Bitcount.

To apply back-propagation through a sign function which is nondifferentiable, the straight-through-estimator (STE) [28] concept is introduced. The function of STE is defined as follows(4)clip(x,−1,1)=max(−1,min(1,x)).

With STE, it is possible to train BNNs directly using the same gradient descent method as normal full-precision neural networks. This sign function acts as an activation function in the network. However, when using the clip function in backpropagation, if the absolute value of the full-precision activation is greater than 1, it cannot be updated in backpropagation. Therefore an identity function is used to approximate the derivative of the sign fnction:(5)$$\frac{\vartheta L}{\vartheta a_{R}}=\frac{\vartheta L}{\vartheta a_{B}}*1_{\left|a_{R}\right|\leq 1},$$
where $a_{R}$ is the real-valued input to the activation function and $a_{B}$ is the binarized output of the activation function. $1_{\left|a_{R}\right|\leq 1}$ is the indicator function that evaluates to 1 if $\left|a_{R}\right|\leq 1$ and 0 otherwise. With this approach, the binary neural network can be used to update the parameters using a powerful back propagation (BP) algorithm based on gradient descent.

### 3.2. Batch Normalization

Batch Normalization (BN) is a technique used to improve the training of deep neural networks. It involves normalizing the activations of each layer in a mini-batch before applying any learnable transformation. Consider a mini-batch B of activations for a given layer, where *m* is the mini-batch size.(6)$$B=\{x_{1\ldots m}\}$$
Let the normalized values be ${\hat{x}}_{1\ldots m}$, and their linear transformations be $y_{1\cdots m}$.(7)$$BN_{\gamma ,\beta }:{\hat{x}}_{1\ldots m}\to y_{1\ldots m}$$ Ioffe et al. [6] present the BN Transform in Algorithm 1. In the algorithm, $\mu_{B}$ is the mini-batch mean; $\sigma_{B}$ is the mini-batch variance; ϵ is a constant added to the mini-batch variance for numerical stability; γ and β are trainable affine transformation parameters (scale and shift).
**Algorithm 1** BN: Batch Normalization**Input:** Values of x over a mini-batch: $B=x_{1\cdots m}$; Parameters to be learned: γ,β**output:** $y_{i}=BN_{\gamma ,\beta }\left(x_{i}\right)$$\;\;\;\;\mu_{B}\leftarrow \frac{1}{m}\sum_{i=1}^{m}x_{i}$$\;\;\;\;\sigma_{B}\leftarrow \frac{1}{m}\overset{m}{\underset{i=1}{\sum }}(x_{i}-\mu_{B}{)}^{2}$$\;\;\;\;\hat{x_{i}}\;\leftarrow \frac{x_{i}-\mu_{B}}{\sqrt{\sigma_{B}^{2}+\varepsilon }}$$\;\;\;\;y_{i}\leftarrow \gamma \hat{x_{i}}+\beta \equiv BN_{\gamma ,\beta }\left(x_{i}\right)$

This Batch Normalization process helps stabilize and accelerate the training of neural networks.

## 4. Method

### 4.1. Attention-Based Batch Normalization

Batch Normalization (BN) plays a crucial role in fitting the distribution of input data by recovering its statistics. In the recovery step, BN learns a pair of per-channel parameters, γ and β, to adjust the mean and variance of each channel.(8)$$y=\gamma \hat{x}+\beta$$

However, using only two parameters to fit the input distribution and recover the representation ability for each channel is challenging, particularly when the input contains complex scenes. This is because in such cases, the statistical data of each channel may be influenced by other channels, and the input data distribution may be more complex and diverse. Therefore, the BN method with only two parameters may not be sufficient to handle such situations, and more complex and flexible methods are needed to deal with complex input data. To address this issue, we propose to use the sigmoid function, which is widely used in attention mechanisms, to enhance the linear transformation of the BN layer and improve its expressive power as follows:(9)$$y=sigmoid\left(\hat{x}\right)\hat{x}+\beta ,$$
where the sigmoid function is defined as:(10)$$sigmoid\left(\hat{x}\right)\;=\;\frac{1}{1+e^{-\hat{x}}}.$$

$\hat{x}$ represents the input feature after normalization. When $\hat{x}$ passes through this sigmoid activation, it undergoes a transformation that confines its output within the desired range of [0,1], which yields the attention score. The output of the sigmoid function can be readily interpreted as the degree of the corresponding pixel being significant or important within the context of the given task. In ABN, we replace the learnable scaling parameter γ in the original BN formulation with an attention score computed as $sigmoid(\hat{x})$, and retain the learnable bias parameter β. As a result, the modulation of $\hat{x}$ becomes input-dependent, introducing nonlinearity into the normalization process.For a detailed description of ABN see Algorithm 2.
**Algorithm 2** ABN: Attention-Based Batch Normalization**Input:** Values of x over a mini-batch: $B=x_{1\cdots m}$; Parameters to be learned: β**output:** $y_{i}=ABN_{\beta }\left(x_{i}\right)$$\;\;\;\;\mu_{B}\leftarrow \frac{1}{m}\sum_{i=1}^{m}x_{i}$$\;\;\;\;\sigma_{B}\leftarrow \frac{1}{m}\overset{m}{\underset{i=1}{\sum }}(x_{i}-\mu_{B}{)}^{2}$$\;\;\;\;\hat{x_{i}}\;\leftarrow \frac{x_{i}-\mu_{B}}{\sqrt{\sigma_{B}^{2}+\varepsilon }}$$\;\;\;\;y_{i}\leftarrow sigmoid\left(\;\hat{x_{i}}\;\right)\hat{x_{i}}+\beta \equiv ABN_{\beta }\left(x_{i}\right)$

### 4.2. Why Does the Attention-Based BatchNorm Work?

The Attention-based BatchNorm (ABN) is the first specialized BN algorithm proposed for BNN our knowledge. In this subsection, we further elaborate on its advantages over existing mechanisms.

#### 4.2.1. Capturing Discriminative Features

To study the ability of ABN to capture and exploit features of a given target, we apply Grad-CAM [29] to compare the regions where different BN methods localize with respect to their target prediction. Grad-CAM is a technique that generates a heatmap to highlight the areas of network attention using the gradient related to the given target. In BNNs, the representational power of the model is somewhat limited by the fact that the weights are restricted to binary values (typically +1 and −1). However, by using Grad-CAM, we can still understand the distribution of attention of a binary neural network on a given image. This allows us to speculate on how the model makes classification decisions and which regions are critical to the classification results.

Figure 1 shows the visualization results of BinaryNet applying the original BN and our ABN on the CIFAR-100 validation set. The red regions indicate the areas of an image that are most important for the network to obtain a high target score, while the blue regions represent the areas that are less relevant for the target score. This suggests that the ABN approach may have a more remarkable ability to highlight each class’s distinctive features than the baseline BN approach. Therefore, it is logical to conclude that ABN provides an additional boost to the final classification performance, as the ability to distinguish between different features is critical for accurate classification. This is further supported by the experimental results of image classification presented in Section 5.

#### 4.2.2. Providing Additional Activation Functions

Typically, the activation function is applied after Batch Normalization to introduce nonlinearity into the network. This enhances the network’s expressive power, allowing it to handle more complex problems than just stacking linear transformations in deep neural networks.

Binary Neural Networks (BNNs) use the STE method to handle the non-differentiability of the Sign function. STE enables approximate gradients in BNNs, facilitating valid backpropagation and efficient training of binary weights. However, a drawback of STE is its impact on traditional activation functions within the network. To enable the use of approximate gradients, binary neural networks commonly adopt a hard Tanh function as the activation function. The hard Tanh function modifies the traditional Tanh function by constraining its output to the range of [−1, 1] and truncating inputs beyond this range, causing the function to have a flat output in this truncated region. This hard restriction leads to the activation function having zero derivatives in that region, making it prone to the gradient vanishing problem during training. As a consequence, the effectiveness of the traditional activation function is weakened.

In contrast to the standard Batch Normalization (BN), our proposed ABN introduces a sigmoid activation into the normalization layer, enabling a data-dependent nonlinear transformation. This design allows the network to better capture complex patterns and intricate relationships within the data. To illustrate the effect of this transformation, we visualize the evolution of activation distributions in Figure 2. The figure shows three stages of the normalization process—raw input $x_{i}$, standardized output ${\hat{x}}_{i}$, and final transformed output $y_{i}$—stacked across training epochs (*Y*-axis). It is evident that the data distribution undergoes a more significant transformation through ABN compared to the linear transformation produced by the baseline BN. Although ${\hat{x}}_{i}$ is normalized to zero mean and unit variance, the actual value range may exceed [−1,+1] due to natural variation in input statistics. The stronger nonlinearity induced by ABN, especially in early training epochs, allows the network to emphasize informative features more effectively, thereby improving convergence stability and performance on complex tasks.

#### 4.2.3. Reshaping the Activation Distribution

In neural networks, activations are dynamic and influenced by the input data. For Binary Neural Networks (BNNs), the distribution of binary activation values is considered a crucial factor that impacts their performance. Existing research has demonstrated that having unbalanced activation values in BNNs can lead to improved accuracy in the network.

Unbalanced activation values refer to situations where the binary activations have an unequal distribution of +1 and −1 values. This non-uniform distribution of activations provides the network with the ability to effectively capture and represent complex patterns in the data. It allows the network to learn more discriminative features and make better decisions when classifying different data samples.

In our experiments, we have observed that our ABN method further increases the imbalance of the binary activation distribution compared to the baseline BN method. Figure 3 illustrates the data distribution of full-precision activations and their binary activations before the binary convolution is computed and summed over the binary activations. The Y-axis in Figure 3 represents the training epoch index, increasing from top to bottom. Each horizontal slice corresponds to the activation distribution at a specific training epoch. The data distribution is most balanced when the absolute value of the sum is zero. However, upon closer examination of the figure, we can observe that the sum of binary activation values gradually decreases during the training process, regardless of whether BN or ABN is used. In other words, the distribution of binary activations becomes more and more unbalanced as the training progresses. By having an unbalanced distribution of activation values, BNNs can exploit the diversity in the data more effectively, enhancing the model’s ability to generalize and improve its performance on various tasks.

## 5. Experiments

In this section, our primary emphasis revolves around one of the fundamental objectives of deep learning—the image classification task. This task stands as a cornerstone in the domain of artificial intelligence, playing a pivotal role in enabling machines to comprehend and interpret visual information. We start by providing a detailed description of the implementation of our experiments (refer to Section 5.1). This allows for a clear understanding of the methodology employed in our study. We then illustrate the tailor-made nature of our ABN for BNNs through a series of experiments (refer to Section 5.2). Subsequently, to assess the impact of the scaling factor γ and bias β in ABN, we conduct an ablation analysis (refer to Section 5.3). Through this analysis, we aim to identify the significance of these components in our proposed method. Finally, we present the results of a comprehensive experiment where we compare our ABN with the baseline BN (Batch Normalization) method, validating the superiority of our proposed approach (refer to Section 5.4).

### 5.1. Implementation Details

To evaluate our Attention-based Batch Normalization (ABN), we conduct extensive experiments on three public benchmark datasets, including CIFAR10 [30], CIFAR100 [30], and TinyImageNet [31], and use standard data augmentation (i.e., random crop and horizontal flip). The details of the datasets and their corresponding experiment setups are given in Table 1.

The BinaryNet, initially proposed by Courbariaux et al. [2], represents the pioneering structure in the realm of BNNs. On the other hand, ResNet-18 [32] introduced the concept of residual blocks, proving highly beneficial in training deeper neural networks. Consequently, many subsequent BNN structures have been built upon the ResNet-18 architecture due to its wide acceptance and success. In this paper, we leverage the prevalence of BinaryNet and ResNet-18 to demonstrate the applicability of ABN in BNNs. In both networks, the weights and activations are binary, with the exception of the first and last layers where full-precision weights are retained. To clarify the proportion of the binary weights in the network, we give in Table 2 the number of binary and full-precision trainable parameters for BinaryNet and ResNet-18 on different datasets.

In the experiments of this paper, we use the baseline BN, which is often used in BNN studies, to conduct a comparative study experiment with our ABN. Our method is implemented in PyTorch [33] on a single NVIDIA 2080Ti GPU and is developed based on the software framework released by BinaryNet’s authors We use the BNN code available at https://github.com/itayhubara/BinaryNet.pytorch (accessed on 8 June 2025).

Our results show that the accuracy obtained by the ABN method on both BinaryNet and ResNet-18 network structures consistently outperforms baseline BN in experiments on these three datasets.

### 5.2. Exclusively Tailored for Binary Neural Networks

To validate the effectiveness of our ABN approach tailored specifically for BNNs, we carried out an experimental comparison between different neural network architectures(binary and full-precision) on the CIFAR-100 dataset. The results of the experiment are shown in Figure 4, where “binary” represents the BinaryNet model, a binary neural network model similar to VGG-11, and “full-precision” represents VGG-11. The two models are cross-compared using BN and ABN methods, respectively.

As shown in the figure, the experimental outcomes for BinaryNet employing the ABN method demonstrate a notable improvement compared to BinaryNet using BN. This observation serves as compelling evidence supporting the effectiveness of our ABN approach when applied specifically to BNNs. Additionally, it is worth noting that in the context of full-precision neural networks, a divergent trend emerges. Specifically, when our ABN method is applied within the full-precision network, it exhibits inferior performance compared to the BN method on the validation set. These experimental results distinctly indicate that our ABN method is tailored exclusively for binary neural networks and does not yield the same benefits when integrated into full-precision neural networks. In fact, its presence within full-precision networks appears to have a detrimental effect on their performance.

In summary, our experiments unequivocally demonstrate that the ABN method significantly enhances the performance of Binary Neural Networks while exhibiting contrasting effects on full-precision networks, where it tends to degrade their performance. This highlights the specificity of the ABN method, underlining its effectiveness as an optimization tool uniquely suited to BNN models.

### 5.3. Ablation Study

In this section, we undertake ablation experiments within the ABN framework to investigate the impact of the scaling factor γ and bias β, which are integral components of the baseline BN method. We perform two distinct experiments, the results of which are shown in Figure 5.

Initially, we introduce the trainable parameter γ into ABN, yet intriguingly, its presence did not yield any discernible enhancement in network performance based on the experimental outcomes. Conversely, when we removed the trainable parameter β from the ABN method, a significant deterioration in network performance was observed. This observation strongly suggests that while γ is dispensable for ABN, β plays a crucial and indispensable role.

In summary, our ablation experiments with ABN provide valuable insights: the absence of a discernible impact from γ implies its non-essential nature within the ABN, while the substantial performance drop upon β removal underscores its necessity for the effectiveness of ABN. We consider the trainable parameter β to be the threshold that determines whether the binary activation value flips between −1 and +1. These findings contribute to a clearer understanding of the intricate dynamics within the ABN method and guide its further refinement and application.

### 5.4. Comparison

We conducted experiments to compare our ABN method with the benchmark BN method using two binary neural network models, BinaryNet and ResNet-18 (https://github.com/itayhubara/BinaryNet.pytorch, accessed on 8 June 2025), across three benchmark datasets: CIFAR-10 and CIFAR-100 (Canadian Institute for Advanced Research, Toronto, ON, Canada), and TinyImageNet (Stanford University, Stanford, CA, USA).

All results (except for TinyImageNet) are averaged over five independent runs with different random seeds to ensure statistical reliability. For TinyImageNet, due to computational constraints, we report results from a representative single run. We did not use k-fold cross-validation; instead, a fixed 10% of the training data was used as a validation set during training, which is a common practice in BNN evaluation. Table 2 lists the parameter scales and validation set classification experimental results of the two models on the three datasets.

From the experimental outcomes, it is evident that our ABN method consistently outperforms the benchmark BN. Moreover, on the CIFAR dataset, our ABN method exhibits the least variance in validation accuracy. This observation suggests that ABN yields more stable classification results, underscoring the algorithm’s enhanced robustness in comparison to the benchmark BN.

## 6. Conclusions

In this paper, we propose a dedicated Batch Normalization method designed specifically for BNNs, i.e., ABN, which achieves competitive advanced performance compared to the most common baseline BN. Specifically, we introduce a self-attention mechanism that rescales the corresponding output channels of the BN according to the existing distribution of activation values and retains trainable bias parameters to provide thresholds for flipping activation values. In addition, for the effective mechanism of ABN action in BNNs we performed a detailed experimental analysis. We identify that the self-attention mechanism in ABN significantly improves its ability to highlight image features; the additional presence of activation-like functions in the method enables the network to capture complex patterns and nonlinear relationships in the data; ABN also amplifies the imbalance in the binary activation distribution, which contributes to improved model generalization and enhanced performance across a range of tasks. With the contribution of these techniques together, the experimental results of our ABN method are significantly improved compared to the baseline BN method on the CIFAR-10, CIFAR-100, and TinyImageNet image classification datasets. Our future work will continue to dig into the deeper mechanisms of BNNs optimization.

## Figures and Tables

**Figure 1 entropy-27-00645-f001:**
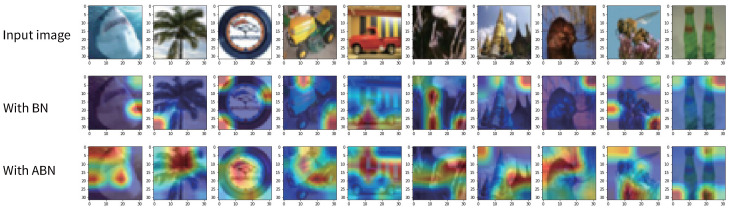
**Visualization of feature activations obtained by different methods.** All comparison methods were trained on CIFAR-100 [30] using the BinaryNet [2] model with consistent settings. The features are extracted on the validation set and shown by Grad-CAM [29]. Our ABN method helps the network to focus on the main regions of the image features.

**Figure 2 entropy-27-00645-f002:**
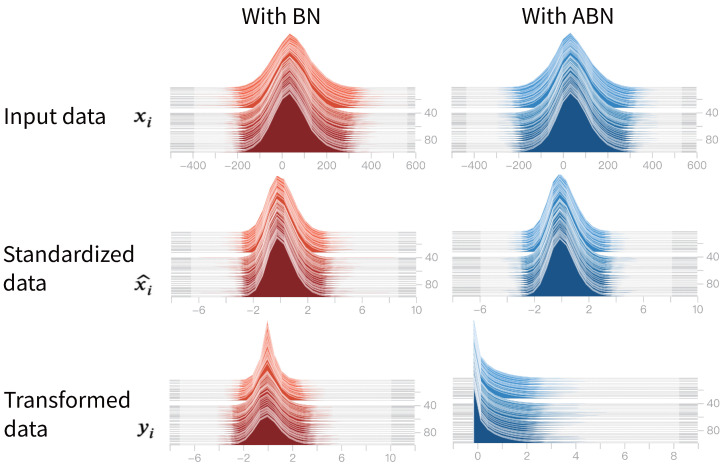
Comparison of changes in data distribution during calculations by baseline BN and ABN methods.

**Figure 3 entropy-27-00645-f003:**
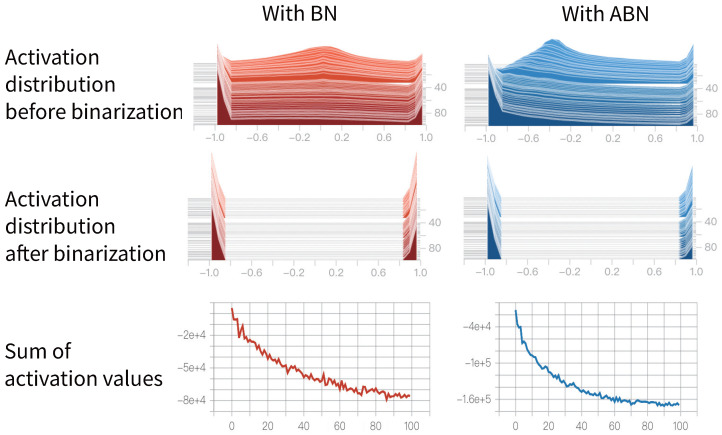
Data distribution of activation values before and after binarization when using different BN methods.

**Figure 4 entropy-27-00645-f004:**
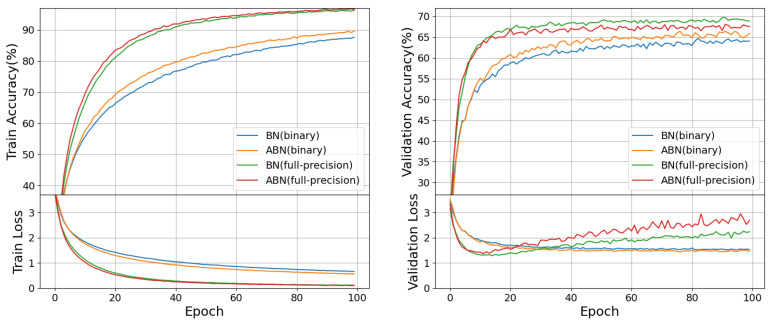
The performance of baseline BN and ABN on BNNs and VGG-11.

**Figure 5 entropy-27-00645-f005:**
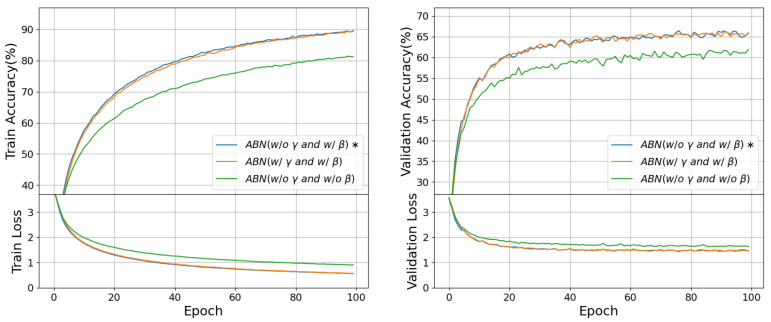
Experiments on the ablation of γ and β in ABN. The * denotes the baseline parameters of ABN, with γ being non-trainable and β being trainable.

**Table 1 entropy-27-00645-t001:** Experiment setup on different datasets.

Dataset	Image	Class	Train/Val	Batch	Epoch
CIFAR-10	32 × 32	10	45 k/5 k	256	200
CIFAR-100	32 × 32	100	45 k/5 k	256	200
TinyImageNet	64 × 64	200	100 k/10 k	256	100

**Table 2 entropy-27-00645-t002:** Comparison of BinaryNet and ResNet-18 using BN and ABN on different datasets. Validation accuracies for different methods. Results are averaged over five runs with different random seeds (except for TinyImageNet). Our ABN method consistently outperforms the baseline BN method. Note: Binary (Binary/All) * refers to the number of binary trainable parameters and their proportion relative to all trainable parameters.

Architecture	Dataset	Binary (Binary/All) *	Accuracy Comparison	
			BN	ABN (Ours)
BinaryNet	CIFAR-10	14,008,320 (99.2%)	89.37±0.10	90.52±0.09
	CIFAR-100	14,008,320 (99.19%)	64.44±0.33	66.10±0.15
	TinyImageNet	39,174,144 (99.45%)	45.10	**47.17**
ResNet-18	CIFAR-10	4,326,400 (99.73%)	87.15±0.31	89.21±0.20
	CIFAR-100	4,326,400 (99.07%)	56.34±0.16	60.43±0.11
	TinyImageNet	4,326,400 (94.19%)	42.35	**45.52**

## Data Availability

The datasets used in this study are publicly available: CIFAR-10 and CIFAR-100 at https://www.cs.toronto.edu/~kriz/cifar.html (accessed on 8 June 2025), and TinyImageNet at https://paperswithcode.com/dataset/tiny-imagenet (accessed on 8 June 2025). Code is available at: https://github.com/gushan/ABN (accessed on 8 June 2025).

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
