# Peer review of "Attention-Based Batch Normalization for Binary Neural Networks"

_entropy, 2025, doi:10.3390/e27060645_

Round 1

Reviewer 1 Report

Comments and Suggestions for Authors

In this paper authors propose the use of the sigmoid function as an attention mechanism during BNN training. To my knowledge, this has not been proposed before. This mechanism is supposed to improve the linear mapping capability in the transformation that is achieved by batch normalization. In essence, the proposed mechanism converts a linear batch normalization into a non-linear one. In addition, the proposed mechanism also increases the imbalance in the activation functions, which improves the learning ability of the model.

In my opinion, the main problem with this paper is that the experiments are not described in detail. I think it is necessary for the authors to describe precisely how the experiments have been performed. For example, with the results shown in section 4, do the results in the graphs refer to the results of a single experiment or are they averaged values from several experiments using, for example, k-fold cross-validation? This is very important to know the significance of the results shown in this work compared with others published.

Other details

The index of the first session, Introduction, is 0, which is unusual.

Figure 2. This figure shows an example of data transformation with the BN and ABN methods. This transformation is highly dependent on the parameter beta (algorithm 2). What value of beta has been used in this transformation? Are the values of beta for the BN and ANB transformation the same or is the process of learning different?

On the other hand, what do the values of the y-axes in the three figures (numbers of 80 and 40) mean? In the graph, the intermediate graph should have its values in [-1,+1], but it is not clear in this figure.

The Y-axis values in figure 3 should also be explained for the same reasons as in figure 2. Also, in the graph "activation distribution after binarization", shouldn't they be discrete values -1 or +1 instead of real values with minimum at -1 and maximum at +1?

In line 113 "The" should be changed to "the".

Reviewer 2 Report

Comments and Suggestions for Authors

In the proposed study, the authors introduced the attention-based batch normalization for Binary Neural Networks, and evaluated its effectiveness for image recognition tasks. The integration of an attention mechanism within batch normalization shows promise, particularly in enhancing robustness and generalization when dealing with noisy data. It is noteworthy that the authors may be the first to apply such a mechanism to BNNs. Additionally, the provision of a code repository is also an advantage of the work. Though, the following improvements are highly recommended prior to publication:

1) Including a baseline model without any batch normalization would provide a clearer demonstration of the proposed method's benefits. Please, add the baseline without batch normalization.
2) Investigating the stability of the approach across a range of batch sizes, both smaller and larger than those currently reported, would offer deeper insights into its performance, given the known impact of batch size on model stability. Please, add the results of testing the methods on other batch sizes.
3) The current literature review containes only a couple of works devoted to batch normalization for Binary Neural Networks (BNNs), with the most recent dated 2021. However, even a preliminary search revealed more relevant and recent studies, including but not limited to: [10.1109/CVPRW53098.2021.00520], [10.3390/s24154780], and [10.1109/ACCESS.2024.3444481]. It is recommended to extend the literature review to incorporate these and other pertinent works. This will ensure the manuscript accurately reflects the current research landscape and clearly positions the contributions of this study within it.

In addition, the following minor revisions would enhance the overall quality of the work:

1) While the code repository is a significant asset, adding comprehensive instructions and detailing the versions of the libraries used would greatly improve its accessibility and usability.
2) To address the visibility issues in Figure 1, it is recommended to incorporate enlarged versions of key components within the main text and to provide an extended version with more examples in the supplementary information. Moreover, including examples where the method encounters difficulties would be beneficial for delineating its scope and identifying potential directions for future improvements.

Reviewer 3 Report

Comments and Suggestions for Authors

Attention-based batch normalization has been an established practice in the field for some time. While specific implementations may vary, the claim that this paper is the first to introduce the approach appears to be somewhat overstated.

Although I’m not completely certain, the method in this paper seems like a special case or a variation of what Liang et al. proposed. Regardless, the authors are encouraged to highlight what is truly novel in their proposed method and how it meaningfully differs from existing attention-based batch normalization techniques.

I have listed two paper below just for references. It would be beneficial for the authors to find additional relevant references , to better position their contribution within the broader context of existing literature.

Liang, S., Huang, Z., Liang, M., & Yang, H. (2020, April). Instance enhancement batch normalization: An adaptive regulator of batch noise. In Proceedings of the AAAI conference on artificial intelligence (Vol. 34, No. 04, pp. 4819-4827).

Martinez, B., Yang, J., Bulat, A., & Tzimiropoulos, G. (2020). Training binary neural networks with real-to-binary convolutions. arXiv preprint arXiv:2003.11535.

Round 2

Reviewer 1 Report

Comments and Suggestions for Authors

The authors of the paper have taken into account all the comments I made in my review, and have improved the quality of the paper by adding and clarifying the necessary content.

Author Response

We thank the reviewer for their time and for the positive evaluation.  
As there were no further comments, no changes were required in the revised manuscript.

Reviewer 2 Report

Comments and Suggestions for Authors

I thank the authors for the work done to improve the manuscript.

The authors have successfully addressed all the issues, except for one point that seems to be quite significant.

The authors mentioned the instability of the training process when training without batch normalization. Therefore, it becomes even more important to check the stability of the developed approach with other batch sizes. Please test the approach with at least a smaller batch size to confirm its stability. In many applied problems, the used batch size is significantly less than 256, and it is necessary to understand whether the proposed method is suitable for these cases.

Reviewer 3 Report

Comments and Suggestions for Authors

issue addressed in authors reply, no further comments 

Author Response

(The authors gave the same response as above.)

Round 3

Reviewer 2 Report

Comments and Suggestions for Authors

Thank you for your work to improve the manuscript!